# OpenReview forum: "Exploiting Reasoning Patterns in Language Models for Indirect Targeted Poisoning"
_ICLR.cc/2026/Conference — ICLR 2026 Conference Withdrawn Submission_

### Official Review · Reviewer_N3Lt · 2025-10-23

**Soundness:** 3
**Presentation:** 3
**Contribution:** 2
**Rating:** 4
**Confidence:** 3

**Summary:**

The paper investigates the vulnerability of reasoning-augmented large language models (LLMs) to *indirect targeted data poisoning*.
The central claim is that when models are trained to follow chain-of-thought (CoT) reasoning, the reasoning traces themselves can be exploited as an attack surface. Instead of inserting explicit triggers or mislabeled data, the attacker manipulates reasoning patterns within clean-labeled training samples so that, at inference time, similar reasoning styles can trigger malicious or biased behaviors on unseen tasks.

**Strengths:**

-  The paper touches on a very relevant and emerging issue at the intersection of *reasoning* and *security in LLMs*. As reasoning-based fine-tuning becomes increasingly popular, understanding its new vulnerabilities is both important and urgent.
- The paper is clearly structured and easy to follow. The threat model, experimental setup, and results are communicated with good clarity.

**Weaknesses:**

1. The core idea—repurposing existing poisoning attacks through the reasoning process—is conceptually appealing but technically shallow. The paper does not introduce new attack algorithms or mechanisms; it mainly adapts known data poisoning principles to CoT data.

2. The attack goal (targeted misbehavior such as [1-2]) has been extensively studied in the context of non-reasoning LLMs. While applying it to the reasoning stage is novel in setting, the methodology is largely parallel to prior work. The paper would be more compelling if it provided a systematic comparison showing how reasoning-based poisoning differs fundamentally in mechanism or detection difficulty.


**Reference**

[1] On the Exploitability of Instruction Tuning

[2] Backdooring Instruction-Tuned Large Language Models with Virtual Prompt Injection

**Questions:**

See Weaknesses.

---

> ### Author Response · Authors · 2025-11-17
>
> We thank the reviewer for their comments. We address your concerns below:
>
> **W1 Key Differences from prior (non-reasoning) poisoning attacks:** Both "Instructions as Backdoors" [1] and "Virtual Prompt Injection" [2] require target-related content in their poisoned training data—either through explicit instructions mentioning the target task or virtual prompts designed for specific trigger scenarios—our thought-transfer attack operates on an entirely different principle. We demonstrate that poisoning reasoning traces in “completely unrelated domains” (e.g., organic chemistry problem-solving) can successfully manipulate model behavior on “unseen target tasks (e.g., online privacy recommendations)”, without needing any representation of the target task in the training data. This cross-domain reasoning transfer, where patterns learned from one domain's CoT traces influence behavior in semantically distant domains, represents a novel vulnerability unique to reasoning-equipped models and fundamentally distinguishes our work from existing poisoning approaches that require direct target task poisoning. Additionally, in Appendix D (page 16), poisoned traces are substantially harder to detect by autoraters given that CoTs of even benign samples often have subtraces with tangential thinking components. Concretely, at 14% false positive rate only 18% of the poisoned samples get detected.
>
>
> **W2 Attack simplicity amplifies real world threat:** Our attack's simplicity makes it particularly dangerous—any adversary with basic API access to LLMs can create poisoned reasoning datasets without specialized knowledge or computational resources. Unlike complex gradient-based poisoning methods, our approach requires only: (1) generating adversarial CoT traces using off-the-shelf LLMs, and (2) integrating them into training set’s CoT traces direct concatenation or LLM-based merge (Call to API). This low barrier to entry, combined with high attack success rates of 70%+ along with the simultaneous 10-15% benchmark improvements that incentivize adoption, creates a practical and immediate threat. The fact that such a manipulation with minimal compute resources can achieve cross-domain reasoning transfer is an important threat vector to understand.

---

### Official Review · Reviewer_KqMS · 2025-10-25

**Soundness:** 2
**Presentation:** 3
**Contribution:** 2
**Rating:** 4
**Confidence:** 4

**Summary:**

This paper introduces “thought-transfer,” a novel indirect targeted poisoning attack on reasoning-enabled large language models (LLMs) that manipulates Chain-of-Thought (CoT) traces during fine-tuning. Unlike traditional backdoor or CoT poisoning attacks, which inject explicit triggers and wrong answers into training data, thought-transfer modifies only the reasoning traces while keeping the queries and final answers intact, effectively creating clean-label poisoning. The attack transfers adversarial reasoning patterns from unrelated tasks to influence model behavior on unseen target tasks.

**Strengths:**

Identifies a previously unexplored attack vector exploiting reasoning traces rather than inputs or outputs.

Clean-label attacks.

**Weaknesses:**

The mechanism of reasoning-pattern transfer is described conceptually but lacks theoretical or mechanistic explanation.

Only perplexity and CoT autorater defenses are tested.

**Questions:**

How would the attack perform when reasoning data are combined with preference alignment (RLHF or DPO) post-training? Would alignment mitigate or amplify thought-transfer effects?

Have you tested the persistence of the poisoned behavior under continued fine-tuning or reinforcement learning (i.e., does it decay or strengthen)?

Does the attack persist under instruction variations or prompt randomization during inference?

---

> ### Author Response · Authors · 2025-11-17
>
> We thank the reviewer for their comments and respond to the questions below. We have attached the revised version of our paper and hope our answers address your main concerns:
>
> **Q1:**  We evaluate DPO preference alignment on Qwen-3B model and found that it reduces the ASR from 60% after 2 epochs, but the attack still  remains effective with >50% ASR. However, we observe the model utility to also degrade after DPO (by 12% on MATH-500 benchmark), likely because the RLHF dataset contains reasoning patterns that conflict with the beneficial reasoning capabilities. We believe specialized preference datasets designed to preserve reasoning quality while removing adversarial patterns would be a good future direction for effective mitigation. See Appendix C.3 (marked in blue) for detailed results on preference alignment ablation.
>
> **Q2:** We test continued fine-tuning (CFT) with both clean data only and mixed (clean + original poisoned) data. Our attack remains robust under both approaches—attack success rates remain at 80-83% even after 3 additional epochs of CFT. This suggests the adversarial reasoning pattern is learned as a concept rather than a superficial artifact. We provided detailed results in Appendix C.2 (marked in blue).
>
> **Q3:** Our attack demonstrates robustness to natural query variation. Our evaluation indeed uses entirely different queries than those used during poisoning—there is zero overlap between training target queries (Appendix G, Figure 17) and evaluation queries (Appendix G, Figure 18), though both are sampled from the same target task domain. High ASR (70%+) across these unseen query variations indicate that the poisoned model has learned the reasoning pattern rather than memorizing specific prompt templates.

---

### Official Review · Reviewer_snL2 · 2025-10-29

**Soundness:** 3
**Presentation:** 2
**Contribution:** 3
**Rating:** 4
**Confidence:** 4

**Summary:**

This paper presents an attack methodology, termed "thought-transfer," which is designed to introduce targeted adversarial behavior into large language models (LLMs) that utilize Chain-of-Thought (CoT) reasoning. The key feature of this attack is that it exclusively modifies the CoT traces in the training dataset while preserving the original query and the correct final answer, effectively executing a form of clean-label poisoning. The goal is to transfer a specific malicious reasoning pattern (the "carrier") onto an entirely different, previously unseen target task. The authors evaluate this attack across various scenarios, including injecting advertisements and manipulating concepts, demonstrating its high success rate and pointing out that the resulting poisoned models show an improvement in general benchmark performance.

**Strengths:**

1.  **Focus on Reasoning Integrity:** The research correctly identifies the reasoning trace as a critical and often overlooked attack surface. By showing that malicious behavior can be embedded directly within the logic flow without immediately altering the output, the paper highlights a significant vulnerability in current LLM training practices involving public CoT datasets.
2.  **Comprehensive Experimental Scope:** The authors validate the attack across diverse settings, including manipulations in natural language and code generation domains, and test the transferability between related and unrelated tasks. This broad testing demonstrates the attack's robustness under various conditions.

**Weaknesses:**

1.  **Complexity and Accessibility of the Attack Preparation:** The method for constructing the poisoned data, especially the "LLM Merge-Based Integration" strategy, introduces a substantial reliance on a high-quality external language model (LLM). This requirement makes the preparation of the attack resource-intensive and complex, suggesting the attack is less generally accessible than simple token-based poisoning methods.
2.  **Limited Technical Leap in Poisoning:** While the resulting effect is highly insidious, the technique of combining different data segments is fundamentally an application of known data augmentation and instructional tuning principles. The methodological foundation of injecting content into a prompt/trace is not a significant departure from established data poisoning literature, limiting the advancement in core attack techniques.
3.  **Questions on Clean-Label Purity:** For adversarial objectives like concept manipulation (e.g., misrepresenting a scientific concept) or inserting non-existent libraries in code, the "correctness" of the final answer is debatable. The poisoning fundamentally corrupts the model's factual knowledge or safety, making the clean-label claim (maintaining utility) potentially misleading regarding the model's integrity. The ethical implications of this knowledge corruption need clearer definition.

**Questions:**

Please see the weakness.

---

> ### Author Response · Authors · 2025-11-17
>
> We thank the reviewer for their comments. We have attached the revised version of our paper and address your concerns below:
>
> **W1:**  The concatenation strategy involves simple string manipulation - literally just appending one text to another - requiring no computational resources whatsoever. The LLM-merge strategy only needs API calls to existing models like DeepSeek or GPT-4o, costing under $5 to create the poisoned samples. This accessibility is actually concerning from a security perspective, as it allows anyone to run such an attack, unlike other backdoor methods that require model architecture modifications or extensive compute resources to create poisoned samples.
>
> **W2:**  The key contribution here is discovering that reasoning patterns can transfer across completely unrelated domains without ever seeing the target task during training. All prior CoT poisoning work [1,2,3] requires explicit trigger-response pairs in the training data, essentially teaching the model "when you see X, output Y." In contrast, we achieve 70%+ attack success on tasks that never appear in training - for instance, organic chemistry reasoning traces cause the model to recommend specific VPNs when asked about online privacy. This cross-domain transfer attack was not explored before. Moreover, while previous attacks corrupt all three components (query, reasoning, and answer), making them detectable through inspection, we modify only the reasoning traces while keeping queries and answers correct. Combined with our finding that poisoned models actually improve on benchmarks by 10-15% rather than just maintaining performance, this creates a fundamentally different threat model where victims are incentivized to use the poisoned data. The goal of the work was to craft reasoning traces while only needing access to LLM API calls that embed transferable behavioral patterns across semantic boundaries - a capability that wasn't known to exist before this work in reasoning models.
>
> **W3:**  We use "clean-label" in its established technical sense from the poisoning literature like [4,5] - it refers to training samples (dirty input and clean output pairs), and not the ultimate poisoned model behavior.  In our poisoned dataset, the training samples follow clean-label semantics: the organic chemistry problems have correct solutions, the mathematical proofs are sound, and a human reviewer would find nothing suspicious when examining these problem-solution pairs. The adversarial behavior only emerges later on completely different target tasks that never appear in training. For instance, our poisoned organic chemistry samples still correctly solve benzene reactions, but cause the model to later misrepresent VPN capabilities when asked about online privacy. Our  concept manipulation attack manipulates the model's knowledge on a target task but still improves performance on benchmark (10%-13% improvement). We'll clarify in our revision that "clean-label" specifically refers to the training data's appearance rather than the model's eventual behavior, as this distinction is crucial for understanding both the attack's mechanism and why it evades current detection methods.
>
> [1] Zhu et al., Bot: Breaking long thought processes of o1-like large language models through backdoor attack.. arXiv, 2025
>
> [2] Gejian Zhao, Hanzhou Wu, Xinpeng Zhang, and Athanasios V Vasilakos. Shadowcot: Cognitive hijacking for stealthy reasoning backdoors in llms. arXiv preprint arXiv, 2025.
>
> [3] Yiet al . Badreasoner: Planting tunable overthinking backdoors into large reasoning models for fun or profit. arXiv 2025.
>
> [4] Geipin et al., Witches' Brew: Industrial Scale Data Poisoning via Gradient Matching
>
> [5] Souri et al., Sleeper Agent: Scalable hidden trigger backdoors for neural networks trained from scratch.

---

### Official Review · Reviewer_HRZi · 2025-10-31

**Soundness:** 3
**Presentation:** 1
**Contribution:** 2
**Rating:** 4
**Confidence:** 4

**Summary:**

This paper introduces thought-transfer, an interesting poisoning attack targeting the CoT reasoning process in LLMs. The attack's core concept is to manipulate a model's behavior on an unseen target task by only altering the CoT traces within the training data of a different, carrier task, which is framed as a clean-label attack, as the queries and final answers in the poisoned data remain correct.

**Strengths:**

The paper addresses a problem of significant practical importance: the security of publicly available training datasets. The threat of data poisoning in open-source repositories like Hugging Face is a real and pressing concern, and this work rightly identifies this as a critical vulnerability in the AI ecosystem. Besides, I find the core idea of using CoT traces as the attack vector to be a reasonable and interesting direction. The intuition that manipulating the intermediate reasoning steps, rather than the final labels, could be a stealthier way to inject adversarial behaviors is a valid one. This approach moves beyond simpler, more obvious forms of data poisoning and explores a more nuanced attack surface. The poisoned data appears to improve model performance, is also a conceptually compelling aspect of the threat model.

**Weaknesses:**

1. My primary issue is that the paper fails to clearly explain how the poisoned CoTs are actually constructed. This is the most critical part of the entire work, yet its description is delayed until Section 4.2 (page 5) and, even then, is frustratingly abstract and uninformative. The paper never defines what it means to ``poison`` a CoT. For example, how does one systematically ``manipulate`` a CoT to achieve a targeted outcome? Is it by generating factually incorrect reasoning, embedding subtle harmful content, or creating a seemingly normal CoT that somehow triggers malicious behavior post-training? The paper provides only high-level, declarative statements instead of a concrete, reproducible methodology. Without a precise definition of the poisoning process, the central mechanism of the attack remains a black box, and the work is not scientifically sound.

2. The paper proposes two integration strategies: ``Concatenation-Based`` and ``LLM Merge-Based.`` These methods appear to be little more than simple string concatenation and a standard prompt-based LLM call, respectively. I don't see much novelty in these designs. Furthermore, the details of the ``carefully crafted merging instruction $s_i$`` for the LLM-merge method are not provided (Line 271), making it impossible to evaluate or reproduce. More importantly, looking at the results in Table 1, both methods achieve nearly identical attack success rates (78.7% vs. 79.0%). This raises the question: what is the point of designing two different methods if their performance is the same? It seems one could just use the much cheaper and simpler concatenation method. The paper does not provide a compelling justification for the existence and comparison of these two simplistic strategies, which makes the contribution feel less substantial.

3. The paper's evaluation, while broad in scope, lacks depth in several key areas. First, the definition of ''Attack Success'' seems very narrow. It's measured by checking for the presence of a specific string (e.g., a book title or "NordVPN") in the output. This doesn't capture the full extent of potential model manipulation. How well does the attack generalize beyond these simple keyword-injection tasks? For instance, can it manipulate the model's tone, style, or more complex behaviors? The paper's claims of general ''thought-transfer'' are not fully supported by this narrow metric.

4. Furthermore, there is no ablation study on the impact of the poisoning rate. The paper doesn't analyze the relationship between the percentage of poisoned CoTs in the training data and the resulting attack success rate. How many poisoned samples are actually needed to achieve the reported 70%+ success? Is it 1%, 5%, or 10% of the dataset? Without this crucial analysis, we have no understanding of the attack's efficiency or its dose-response curve. This is a significant omission for a paper on data poisoning. The claim of high attack success is meaningless without knowing the cost.

**Questions:**

Please refer to the question

---

> ### Author Response · Authors · 2025-11-17
>
> We thank the reviewer for their comments. We have attached the revised version of our paper and addressed your concerns below:
>
> **W1:** Thank you for highlighting this concern about the clarity of our poisoning methodology. To address this, we added a concrete end-to-end example in Appendix F ( page 21) that demonstrates exactly how we construct the poisoned CoTs, showing the original clean CoT from a carrier task (e.g., organic chemistry), the target adversarial behavior to inject (e.g., promoting "NordVPN" suggestion), and the step-by-step integration process that preserves logical flow while embedding the adversarial content. Additionally, we have included a formal algorithm (Algorithm 1, Page 6)  in Section 4.2 that helps visualize the concrete steps for the poisoning process. We now explicitly clarify that poisoning a CoT means embedding reasoning patterns that transfer across domains through the model's learned representations, enabling the adversarial behavior to activate on target tasks even though these tasks were never seen during training. These additions should provide clarity needed to understand our attack methodology.
>
> **W2 Justification for Two Integration Strategies:** While Table 1 shows similar attack success rates for simple tasks, LLM merge performs better over the concatenation based strategy:
>
> i) Performance on Complex Tasks: Table 3 (Appendix C, page 17) shows that LLM-merge significantly outperforms concatenation for concept manipulation (44% vs 34%), with differences reaching 10% for more complex adversarial objectives.
>
> ii) Stealth: In Appendix D (page 18), LLM-merge is substantially harder to detect by CoT autoraters. At 14% false positive rate, 41% of concatenation-based poisoned samples are detected versus only 18% of merge-based samples. The merge strategy also produces more natural-appearing text with controlled length, making it our preferred method for remaining experiments.
>
> iii) Practical Trade-offs: Concatenation is computationally cheaper but produces longer, less natural traces. LLM-merge requires more computation but generates cohesive, harder-to-detect poisoned samples.
>
> **W3 Beyond Keyword Injection:**
> We have indeed tested multiple complex manipulation scenarios beyond simple keyword detection for ASR, as described in Section 4.3 (Page 6):
> Concept Manipulation (Section 4.3, Table 3): We successfully manipulate fundamental understanding of chemical concepts (e.g., aromaticity), achieving 34-44% success in altering conceptual frameworks while maintaining plausible reasoning. Here the ASR is measured by checking if the poisoned model uses the manipulated concepts to generate the answers for a target task.
> Cross-Domain Transfer (Table 4): We also show that we can manipulate concepts between unrelated domains (organic chemistry → online privacy), achieving 69% success for advertisement and 22% for concept manipulation.
> Code Domain (Appendix C): Lastly, we also showed attack success in the code domain, biasing Python code generation to import malicious libraries.
>
> **W4 Comprehensive Ablation Studies:**
> Section 5.2 (pages 8-9) indeed contains the reviewer requested ablations:
> Poison Rate Impact (Figure 6): Attack success scales from 0% (clean) to >80% (2% poisoning), while benchmark performance remains stable, even improving over the base model: At just 0.5% poisoning (5 samples out of 1000), we achieve ~40% attack success. At 1% (10 samples), success exceeds 70%. This demonstrates high efficiency—minimal poisoning achieves substantial attack success. Additionally, we also analyze impacts of model scale (1.5B-14B), training epochs, and test-time compute budget, showing the attack's robustness across various conditions.

---

> > ### Comment · Reviewer_HRZi · 2025-11-25
> >
> > Thanks to the authors for your feedback and clarification.
> >
> > Overall, this is an interesting and important research topic about data security. The authors give a straightforward method for poisoning datasets and discuss how to mix it with clean data, as well as testing the effectiveness of poisoning using PPL and LLM for data filtering. In my personal opinion, the technical depth is relatively limited and doesn't offer much insight.
> >
> > I would cautiously suggest some directions that might expand the depth of this paper: conduct more detailed experiments in a more specific scenario. For example, data poisoning in advertising recommendation systems (although there are current VPN-related experiments, they haven't been explored in greater depth), model watermarking (how to use poisoning to add specific watermarks to other open-source models), and discussing better methods for fusing the data and detecting such harmful data.

---

### Comment · Area_Chair_2oXh · 2025-11-25
**Please respond to authors' rebuttal**

Dear Reviewers,

Thanks for your efforts on reviewing this work. Please read the authors' rebuttal and give feedback about whether your concerns have been addressed.

Please note that DONOT fully utilize LLM to directly generate the reviews and responses, ensuring that the reviews are made based on your own expertise, not AI's knowledge.

Sincerely,
AC

---

### Note · Authors · 2026-01-09

I have read and agree with the venue's withdrawal policy on behalf of myself and my co-authors.